# Depressive symptoms, functional impairment, and health-related quality of life in idiopathic normal pressure hydrocephalus: A population-based study

**Johanna Andersson**[1]*, **Martin Maripuu**[1], **Mathilda Sjövill**[1], **Anna Lindam**[2],
**Katarina Laurell**[3]

1 Department of Clinical Sciences, Umeå University, Umeå, Sweden, 2 Department of Public Health and Clinical Medicine, Unit of Research, Education and Development- Östersund, Umeå University, Umeå, Sweden, 3 Department of Biomedical and Clinical Sciences, Neurology, Linköping University, Linköping, Sweden

* l.johanna.u@gmail.com

**Data Availability Statement:** Data cannot be shared publicly because the data contains sensitive personal data. Public deposition would breach

## Abstract

### Background

Maximising quality of life is a central goal for all healthcare, especially when dealing with dementing disorders. In this study we aimed to compare health-related quality of life (HRQoL), depressive symptoms and functional impairment between individuals with and without idiopathic normal pressure hydrocephalus (iNPH) from the general population.

### Methods

A total of 122 individuals, 30 with iNPH (median age 75 years, 67 females) underwent neurological examinations and computed tomography of the brain with standardised rating of imaging findings and clinical symptoms. The participants completed the Geriatric Depression Scale (GDS-15) and the HRQoL instrument EQ5D-5L. In addition, the modified Rankin Scale (mRS) was used to evaluate functional impairment.

### Results

Compared with participants without iNPH, those with iNPH reported a higher score on GDS-15 (median 3 *vs* 1) and mRS (median 2 *vs* 1) (p < 0.05). Further, those with iNPH rated lower on EQ5D-5L (index 0.79, VAS 70) than those without iNPH (index 0.86, VAS 80) (p < 0.05). In logistic regression models, low HRQoL was associated with more depressive symptoms, a higher degree of iNPH symptoms, and lower functional status.

### Conclusions

In this population-based sample, those with iNPH had more depressive symptoms, lower functional status, and worse quality of life compared to those without iNPH. The strongest association with low HRQoL was found for depressive symptoms, functional level, and

compliance with the protocol approved by the regional research ethics board. In the case that a researcher or authority would like to access more information, please contact Data Protection Officer (DPO) Sanna Othman, lawyer at the Region of Jämtland -Härjedalen. E-mail address: sanna.othman@regionjh.se Phone number: +46 63 147586 Postal address: Region Jämtland Härjedalen, Box 654, 831 27 Östersund, Sweden.

**Funding:** KL, JA and MM received governmental funding from Region Jämtland Härjedalen. JA received funding from the donation funds of "Syskonen Perssons donationsfond", "Jämtlands cancer- och omvårdnads fond" and "Stiftelsen Forskningsfonden för klinisk neurovetenskap vid NUS". The funders had no role in study design, data collection and analysis, decision to publish, or preparation of the manuscript.

**Competing interests:** The authors have declared that no competing interests exist.

degree of iNPH symptoms. These results underline the value of shunt surgery because of its potential to reduce symptoms and disability in iNPH and therefore improve HRQoL.

## Introduction

Idiopathic normal pressure hydrocephalus (iNPH) is considered a treatable cause of dementia with a prevalence between 0.5 and 3.7% in the age group 65 years and older [1,2]. The condition is characterised by ventriculomegaly, gait impairment, cognitive decline, and urinary incontinence [3]. The only available treatment is shunt surgery, which has an improvement rate of up to 82% [4]. According to outcome data from the Swedish hydrocephalus quality registry, 39% of iNPH patients improved at least one step on the modified Rankin scale (mRS) after shunt surgery [5].

The cognitive symptoms of iNPH include apathy, frontal-subcortical executive dysfunction, and mental slowness [6,7]. Similar symptoms are reported in depressive disorders, which is a common cause of cognitive decline in higher age [8]. Furthermore, depression can coexist with iNPH, possibly worsening the intellectual impairment.

Rates of depression ranges from 14 to 47% in hospital-based materials of patients with iNPH [9–11]. Suspected depression was still more common in iNPH patients (46%) 1–3 years after shunt surgery compared to age- and sex-matched healthy controls (13%) [10]. However, in a study from Japan, the prevalence of depression decreased from 36% before surgery to 5% three months after surgery [12].

Most outcome studies of iNPH focus on improvement in gait and functional level rather than depression, quality of life, and other self-rated outcome measures. Nevertheless, it has been emphasised that maximising quality of life (QoL) is a central goal for all healthcare, especially when dealing with dementing disorders [13].

Junkkari et al. [11] reported health-related quality of life (HRQoL) to be severely affected in iNPH patients and showed that the severity of iNPH as well as the degree of depressive symptoms were associated with a lower HRQoL. In a Swedish study of 118 iNPH patients who underwent shunt surgery, a positive correlation was seen between higher QoL and walking ability [14]. In another study from Sweden, 86% of patients improved HRQoL six months after surgery, almost to the same level as an aged-matched control group [15]. Three months after shunt surgery 30 iNPH patients improved in the EQ5D domains of mobility, anxiety and self-care to the same levels as matched controls [16].

Since most previous studies on iNPH are hospital-based, this study aimed to assess depressive symptoms, functional impairment, and HRQoL in individuals with and without iNPH from the general population. Another aim was to investigate which factors, including clinical and radiological findings, influence low HRQoL.

## Methods

### Study participants

The sample in the present study participated in a two-year follow-up in 2017 of a previously published population-based prevalence study of iNPH [2]. The participants derived from a randomly selected sample of 1000 residents 65 years and older living in the region of Jämtland-Härjedalen, Sweden. Those who reported gait or balance impairment together with one of the other typical iNPH symptoms in a questionnaire were invited together with a subgroup without symptoms for further clinical and neuroradiological investigations. Exclusion criteria

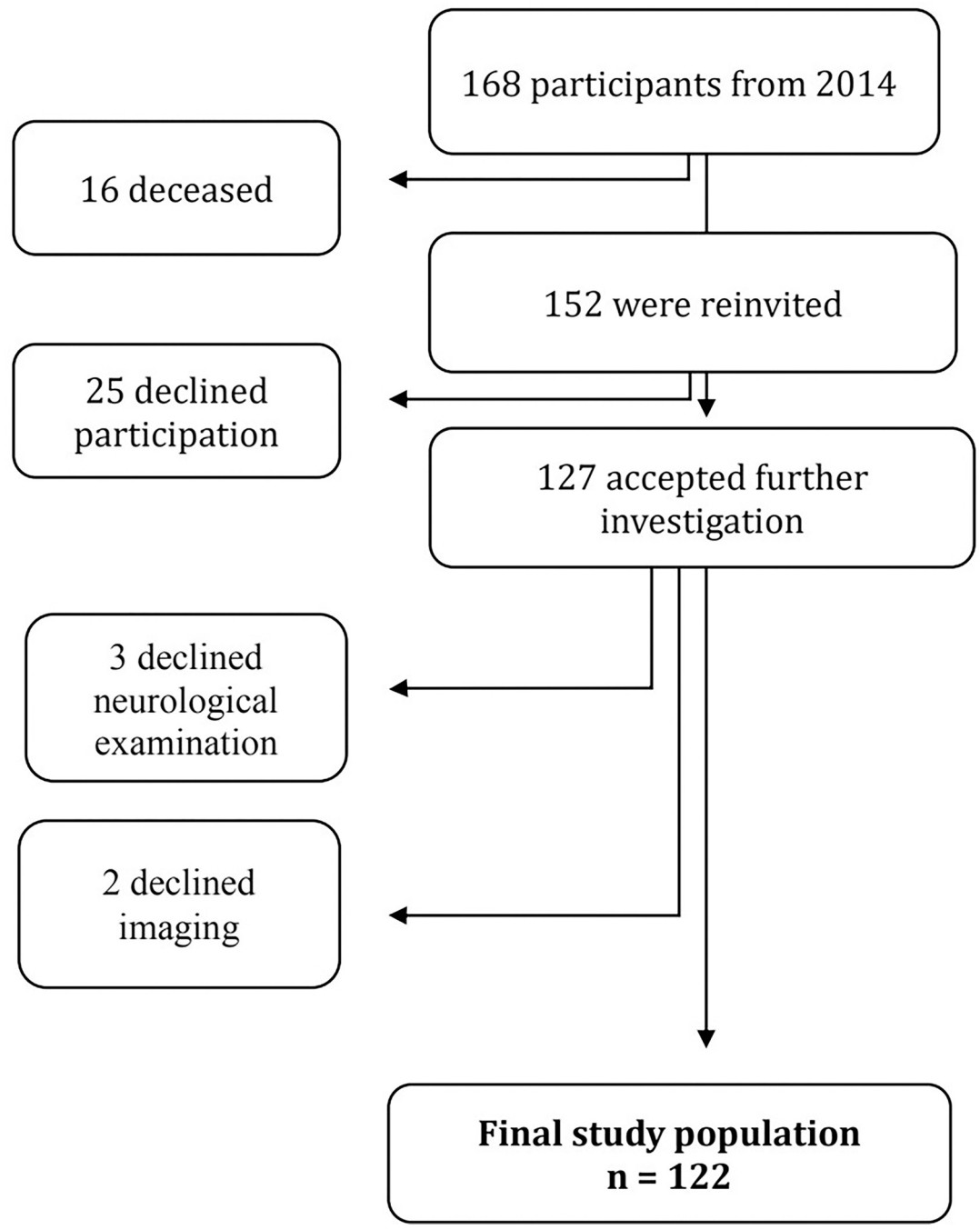

**Fig 1. Flowchart of study participants.**

were severe medical conditions sufficient to explain the symptoms e.g. a known brain tumour, stroke, or traumatic brain injury. A history of mental illness or current depressive disorder was not considered an exclusion criterion, neither in the original study, nor in the follow-up. Of the 168 individuals examined at baseline, 16 were diseased in 2017, so only 152 of the original 168 were invited to the follow-up. Of these 152, 25 declined the invitation and 5 declined either the clinical or radiological examinations, resulting in a final sample of 122 individuals (Fig 1).

**Table 1. Diagnostic criteria.**

| Japanese guidelines 2[nd] edition [19] | | | |
|---|---|---|---|
| | **Possible with MRI support** [a] | **Possible** | **Unlikely** |
| **Clinical** | ≥ 2 symptoms: | ≥ 2 symptoms: | None of the |
| | Gait disturbance | Gait disturbance | symptoms |
| | Cognitive impairment | Cognitive impairment | |
| | Urinary incontinence | Urinary incontinence | |
| **Imaging** | EI >0.3 Narrowing of the sulci over the high convexity/ DESH | EI >0.3 | EI ≤ 0.3 |

Abbreviations: EI = Evans' index, MRI = magnetic resonance imaging, DESH = disproportionally enlarged sulci hydrocephalus.

The ethical committee at Umeå University approved the study in 2014 (Dnr 2014/180-31 and Dnr 2017-167-32 M). The study was also approved by the Radiation Protection Committee (2017-04-24). All participants gave written informed consent.

### Clinical and radiological evaluation

All study participants underwent a neurological examination and testing according to the iNPH symptom scale developed by Hellström et al. [17]. The scale consists of four domains: gait (10-m walking test and ordinal scale), balance (ordinal scale), urinary symptoms (ordinal scale), and neuropsychology (RAVLT, Stroop color and Stroop interference test). The scores range between 0 and 100, where lower scores correspond to more symptoms.

A computed tomography (CT) scan of the brain was conducted within 6 months after the baseline and follow-up clinical examinations (2017). Standardised assessment of the imaging findings was performed with the iNPH Radscale, ranging between 0 and 12, with higher scores representing more radiological signs of iNPH [18]. Typical imaging findings of iNPH are displayed in the Supporting information section (S1 and S2 Figs).

The Japanese diagnostic guidelines [19] were used to diagnose iNPH because these diagnostic criteria have previously shown a better concordance with the clinical evaluation of a senior neurologist compared to the American-European Guidelines [20]. In the present study, the iNPH group included "possible" or "possible with MRI (or CT) support" (see Table 1), whereas the "unlikely iNPH" group did not fulfil these criteria. [19]. Additional detailed descriptions of the diagnostic procedure are provided in the preceding prevalence study [2].

### Evaluation of depressive symptoms

At the two-year follow-up, the participants completed the Geriatric Depression Scale 15 (GDS-15), a screening instrument for depression in the elderly with a sensitivity of 82–100% and specificity of 72–82% [21]. The GDS-15 consists of 15 'yes' or 'no' self-completing questions; a score of ≥ 5 indicates suspected depression [22].

### Evaluation of functional impairment

Modified Rankin Scale (mRS) was used to rate functional impairment [23]. The scale consists of five levels: 0 ('no symptoms') to 5 ('severe disability requiring constant nursing care').

Disability was defined as a mRS score of $\geq$ 2. The rating was performed during the patient interviews at the same visit as the clinical examinations.

## Self-reported health-related quality of life

To investigate HRQoL, the EuroQol five-dimensions 5 levels instrument (EQ5D-5L) was used. Permission to use the instrument was obtained from the EuroQol Research Foundation. The questionnaire is not disease specific and consists of questions on mobility, self-care, usual activities, pain/discomfort, and anxiety/depression. Each question has five alternatives: no problems, slight problems, moderate problems, severe problems, and extreme problems/unable to perform [24]. The result can be converted to an index value, originating from data generated from a general population survey using the time-trade-off method. A value of 1 is considered full health, a value of 0 is considered dead, and a value below 0 is considered a health condition considered by the general population as a health state worse than death. The UK index tariff was used to calculate the index value as there is no Swedish index tariff [25]. EQ5D-5L also consists of a self-rating section with a visual analogue scale (EQ-VAS), 0–100, where 100 is the best health imaginable. To define low HRQoL, we used the lowest quartile of EQ5D-5L index (range: −0.01–0.76) and EQ-VAS (range 8–60 mm) in the total sample as there is no established cut-off level.

The questionnaires regarding HRQoL and depressive symptoms were completed at the same time as the clinical examination. If missing or erroneous, new questionnaires were sent home within 6 months.

## Statistical methods

Because the data were not normally distributed, the descriptive statistics were presented with medians and inter quartile range (IQR). Differences between groups (iNPH vs unlikely iNPH) were analysed with the Mann Whitney U-test for ordinal data and with Pearson chi-squared test or the Fisher's exact test (when the expected number in a subcategory was < 5) for categorical data (Tables 2 and 4).

iNPH diagnosis, age and sex were analysed as risk factor for suspected depression (GDS-15 $\geq$ 5), disability (mRS $\geq$ 2), low EQ5D-5L index, and low EQ-VAS, separately and with univariate and multiple logistic regression models, Odds ratios (OR) and 95% confidence intervals (CI) were estimated. Age, sex, and iNPH diagnosis were independent factors in all analyses (Table 3).

In the next step (Tables 5 and 6), we utilized multiple logistic regression models to investigate the relationship between low HRQoL (low EQ-VAS and low EQ5D-5L index) and several clinically significant parameters such as iNPH symptoms and depressive symptoms. We started with univariate models and then progressed to more comprehensive multiple regression models. Model A included all parameters found to be significant in the univariate analyses, but also adjusted for age and sex.

Models B and C were chosen based on clinical reasoning. Recognizing potential interdependencies and overlap among some independent factors in Model A, we introduced two additional models. Model B, alongside age and sex, incorporated imaging signs (Radscale score) and iNPH symptoms (iNPH score)—factors specific to iNPH. Conversely, Model C included depressive symptoms (GDS-15) and functional impairment (mRS), which are more general factors potentially associated with diminished quality of life. We anticipated that this approach would offer valuable insights by distinguishing between analyses based on distinct sets of predictors.

**Table 2. Demographics and characteristics of participants, Jämtland-Härjedalen, Sweden (n = 122).**

| | Total sample n = 122 | iNPH n = 30 | Unlikely iNPH n = 92 | P-value iNPH vs unlikely |
|---|---|---|---|---|
| Age, Md (IQR) | 75.0 (72.0–79.3) | 80.0 (73.0–83.0) | 74.0 (71.3–77.8) | **0.005**[a] |
| Sex, Female n (%) (n) | 67 (54.9) | 13 (43.3) | 54 (58.7) | 0.142[b] |
| iNPH symptoms | | | | |
| iNPH total score, Md (IQR) | 85.2 (74.0–91.9) | 71.4 (60.8–78.2) | 88.6 (79.3–94.5) | **0.000**[a] |
| Gait domain, Md (IQR) | 90.0 (76.6–100) | 74.5 (57.4–85.3) | 94.3 (84.1–100) | **0.000**[a] |
| Balance domain, Md (IQR) | 83.0 (83.0–100) | 83.0 (67.0–83.0) | 83.0 (83.0–100) | **0.000**[a] |
| Continence domain, Md (IQR) | 80.0 (60.0–100) | 80.0 (55.0–80.0) | 80.0 (60.0–100) | **0.006**[a] |
| Neuropsychology, Md (IQR) | 82.5 (70.0–92.5) | 65.0 (49.4–85.0) | 85.0 (72.5–94.4) | **0.000**[a] |
| MMSE, Md (IQR) | 27.0 (25.0–28.0) | 26.5 (22.8–28.0) | 27.0 (26.0–28.0) | 0.094[a] |
| Imaging signs | | | | |
| Radscale score, Md (IQR) | 2.0 (2.0–4.0) | 4.0 (3.0–5.2) | 2.0 (1.0–3.0) | [a] |

*Note*: Bold values indicate p-values below 0.05.

[a] Mann-Whitney U.

[b] Pearson Chi Square.

One missing observation for MMSE.

Abbreviations: iNPH: Idiopathic normal pressure hydrocephalus; IQR: Inter quartile range; Md: Median.

iNPH score range 0–100 (low score more symptoms).

Radscale score range 0–12 (high score more radiological signs).

The level of significance was set to < 0.05 for all analyses. The analyses were performed using SPSS (version 26).

## Results

The sample consisted of 122 individuals– 67 women and 55 men with a median age of 75 years (range 69–93). The majority of the participants (n = 92) were younger than 80 years old, and the remaining (n = 30) were 80 years of age or older. The criteria for iNPH [19] were fulfilled in 30 individuals, of which 3 were already shunt operated. The remaining 92 were classified as 'unlikely' iNPH. Descriptive data for the two groups are presented in Table 2.

For individuals with iNPH, the likelihood of having disability (mRS scores ≥ 2), low EQ-VAS and low EQ5D-5L index was 3.5, 3.1, and 3.9 times higher, respectively, compared to those without iNPH (figures adjusted for age and sex) (Table 3). The odds for suspected depression (GDS-15 ≥ 5) were not increased in relation to the diagnosis of iNPH (Table 3).

Individuals with iNPH had a higher GDS-15 score (median 3.0, IQR 1.0–5.2) compared to those without iNPH (median 1, IQR 0.0–2.0) (p < 0.01), but the differences in proportions of suspected depression did not reach statistical significance (Table 4, Fig 2). Similarly, those with iNPH reported a functional level (median 2.0, IQR 1.0–2.0) lower than those without iNPH (median 1.0, IQR 0.0–2.0) (p < 0.01), and the proportion with disability was more than twice as high for those with iNPH (Table 4).

In addition, iNPH diagnosis was associated with a lower HRQoL. The EQ5D-5L index was lower for those with iNPH (median 0.79) than for those without (median 0.86) (p < 0.01). Similarly, those with iNPH reported lower values on the EQ-VAS scale (median 70 mm) compared to those with unlikely iNPH (median 80 mm) (p = 0.000) (Table 4 and Fig 2).

**Table 3. iNPH diagnosis, age and sex and the likelihood of depressive symptoms, disability and low HRQoL aseessed with logistic regression.**

| | Univariate logistic regression models | | Multiple logistic regression models | |
|---|---|---|---|---|
| **Dependent factor: GDS-15 $\geq$ 5** | | | | |
| *Independent factors* | **OR (CI)** | **P value** | **OR (CI)** | **P value** |
| Age | 1.06 (0.98–1.15) | 0.181 | 1.04 (0.96–1.13) | 0.355 |
| Sex (male reference) | 1.12 (0.43–2.88) | 0.822 | 1.27 (0.48–3.38) | 0.636 |
| iNPH diagnosis (unlikely iNPH reference) | 2.21 (0.81–6.00) | 0.120 | 1.98 (0.68–5.77) | 0.208 |
| **Dependent factor: mRS $\geq$ 2** | | | | |
| *Independent factors* | | | | |
| Age | 1.13 (1.05–1.21) | **0.001** | 1.10 (1.02–1.19) | **0.012** |
| Sex (male reference) | 1.04 (0.50–2.18) | 0.914 | 1.29 (0.57–2.93) | 0.542 |
| iNPH diagnosis (unlikely iNPH reference) | 4.39 (1.84–10.47) | **0.001** | 3.51 (1.39–8.84) | **0.008** |
| **Dependent factor: Low EQ-VAS[a]** | | | | |
| *Independent factors* | | | | |
| Age | 1.03 (0.96–1.11) | 0.427 | 1.00 (0.93–1.08) | 0.949 |
| Sex (male reference) | 0.85 (0.37–1.95) | 0.704 | 0.96 (0.41–2.27) | 0.928 |
| iNPH diagnosis (unlikely iNPH reference) | 3.15 (1.29–7.71) | **0.012** | 3.10 (1.21–7.98) | **0.019** |
| **Dependent factor: Low EQ5D-5L index[a]** | | | | |
| *Independent factors* | | | | |
| Age | 1.02 (0.95–1.10) | 0.642 | 0.99 (0.91–1.07) | 0.744 |
| Sex (male reference) | 1.30 (0.56–3.02) | 0.548 | 1.56 (0.64–3.81) | 0.333 |
| iNPH diagnosis (unlikely iNPH reference) | 3.49 (1.42–8.60) | **0.007** | 3.94 (1.50–10.38) | **0.005** |

*Note*: Bold values indicate p-values below 0.05.

[a] Low EQ5D-5L: Lowest quartile of EQ5D-5L index (range: −0.01–0.76), Low EQ-VAS: Lowest quartile of EQ-VAS (range 8–60 mm).

Abbreviations: EQ5D-5L EuroQol five-dimensions 5 levels instrument; GDS-15: Geriatric depression scale-15; iNPH: Idiopathic normal pressure hydrocephalus; mRS modified Rankin Scale; VAS visual analogue scale.

GDS-15 range 0–15 (high score more depressive symptoms).

VAS range 0–100 (high score higher self-rated health).

In a univariate logistic regression model with low EQ-VAS as the dependent factor, a significant association was found with the degree of iNPH symptoms, both for the total iNPH score as well as the score for the domains gait, balance, and neuropsychology (Table 5). In addition, a significant association was found for the degree of imaging findings, depressive symptoms, and functional impairment.

When these significant factors in univariate analyses were put together with age and sex in a multiple logistic regression model only depressive symptoms (GDS-15) remained significantly associated with low EQ-VAS (Model A, Table 5). When omitting depressive symptoms and functional impairment (mRS) to study the other variables, only iNPH symptoms (total iNPH score) remained significantly associated with low EQ-VAS (Model B, Table 5). In a third model, both depressive symptoms and mRS were independently associated with low EQ-VAS (Model C, Table 5).

When using low EQ5D-5L index as the dependent factor, the results were similar as the results for low EQ-VAS except for a significant association with continence symptoms in univariate logistic regression (Table 6). In addition, when using the logistic regression models A, B, C as above, depressive symptoms did not reach significant association with low EQ5D-5L index as the dependent factor (Model A, Table 6).

**Table 4. Depressive symptoms, functional impairment and HRQoL among patients with a iNPH diagnosis compared to those unlikely to have a diagnosis.**

| | Total sample n = 122 | iNPH n = 30 | Unlikely iNPH n = 92 | P-value iNPH vs unlikely |
|---|---|---|---|---|
| Depressive symptoms | | | | |
| GDS-15 median (IQR) | 1.0 (0.0–3.0) | 3.0 (1.0–5.2) | 1.0 (0.0–2.0) | **0.002**[a] |
| GDS-15≥ 5 n (%) | 21 (17.2) | 8 (26.7) | 13 (14.1) | 0.114[b] |
| Functional impairment | | | | |
| mRS median (IQR) | 1.0 (0.0–2.0) | 2.0 (1.0–2.0) | 1.0 (0.0–2.0) | **0.001**[a] |
| mRS ≥ 2 n (%) | 45 (36.9) | 19 (63.3) | 26 (28.3) | **0.001**[b] |
| HRQoL | | | | |
| EQ5D-5L index median (IQR) | 0.86 (0.76–0.94) | 0.79 (0.68–0.86) | 0.86 (0.80–0.94) | **0.001**[a] |
| Low EQ5D-5L index n (%)[c] | 29 (24.4) | 13 (43.3) | 16 (18.0) | **0.005**[b] |
| EQ-VAS median (IQR) | 80 (60–89) | 70 (50–75) | 80 (70–90) | **0.000**[a] |
| Low EQ-VAS n (%)[c] | 30 (25.6) | 13 (43.3) | 17 (19.5) | **0.010**[b] |

*Note*: Bold values indicate p-values below 0.05.

[a] Mann-Whitney U.

[b] Pearson Chi Square.

[c] Low EQ5D-5L: Lowest quartile of EQ5D-5L index (range: −0.01–0.76), Low EQ-VAS: Lowest quartile of EQ-VAS (range 8–60 mm).

Three missing observations for EQ5D 5L index and five missing observations for EQ-VAS.

Abbreviations: EQ5D-5L: EuroQol five-dimensions 5 levels instrument; GDS-15: Geriatric Depression Scale 15; iNPH: Idiopathic normal pressure hydrocephalus; mRS: modified Rankin Scale; VAS: visual analogue scale.

GDS-15 range 0–15 (high score more depressive symptoms).

mRS range 0–6 (high score more disability).

VAS range 0–100 (high score higher self-rated health).

## Discussion

Depressive symptoms, functional impairment, and HRQoL in iNPH have previously only been evaluated in clinical materials. In this population-based sample, participants with iNPH had a higher degree of depressive symptoms, lower functional level, and lower HRQoL compared to those with unlikely iNPH. The strongest independent association with low HRQoL was found for depressive symptoms, decreased functional level, and a higher degree of iNPH symptoms.

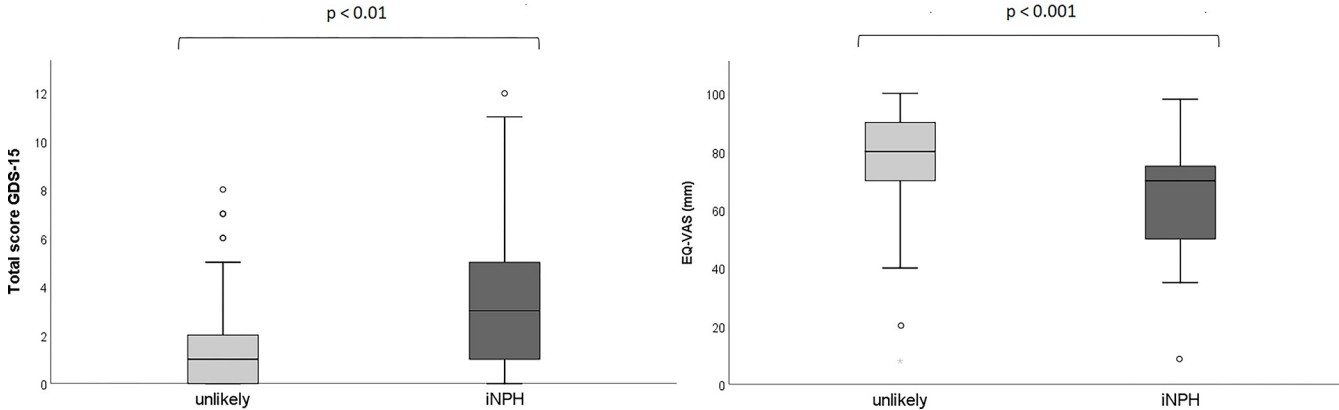

**Fig 2. Difference in depressive symptoms (GDS-15) and self-rated health (EQ-VAS) between unlikely iNPH and iNPH.** EQ-VAS: EuroQol visual analogue scale; GDS-15: Geriatric depression scale-15; iNPH: Idiopathic normal pressure hydrocephalus. Significance level tested with Mann-Whitney-U test.

**Table 5. Low EQ-VAS (lowest quartile of EQ-VAS) in relation to clinical findings, depressive symptoms, and radiological signs of iNPH.**

| Independent factors | Univariate logistic regression models (n = 117) | | Multiple logistic regression models (n = 117) | | | | | |
| --- | --- | --- | --- | --- | --- | --- | --- | --- |
| | | | Model A age, sex, Radscale, iNPH score, mRS and GDS-15 | | Model B age, sex, Radscale and iNPH score | | Model C age, sex, mRS and GDS-15 | |
| | OR (CI) | P value | OR (CI) | P value | OR (CI) | | OR (CI) | P value |
| Age | 1.03 (0.96–1.11) | 0.427 | 0.95 (0.86–1.04) | 0.268 | 0.96 (0.87–1.04) | 0.305 | 0.96 (0.88–1.05) | 0.368 |
| Sex (male reference) | 0.85 (0.37–1.95) | 0.704 | 0.74 (0.27–2.00) | 0.554 | 0.86 (0.34–2.16) | 0.749 | 0.74 (0.28–1.95) | 0.540 |
| Radscale score | 1.26 (1.03–1.53) | **0.023** | 1.03 (0.79–1.33) | 0.839 | 1.09 (0.87–1.38) | 0.442 | | |
| modified Rankin Scale | 2.24 (1.47–3.42) | **0.000** | 1.46 (0.75–2.81) | 0.265 | | | 1.73 (1.04–2.89) | **0.036** |
| GDS-15 | 1.56 (1.27–1.92) | **0.000** | 1.39 (1.10–1.76) | **0.006** | | | 1.41 (1.13–1.77) | **0.003** |
| Total iNPH score | 0.95 (0.92–0.98) | **0.000** | 0.98 (0.93–1.03) | 0.454 | 0.94 (0.91–0.98) | 0.002 | | |
| Domain scores | | | | | | | | |
| Gait | 0.97 (0.95–0.99) | **0.001** | | | | | | |
| Balance | 0.93 (0.90–0.97) | **0.000** | | | | | | |
| Continence | 0.99 (0.98–1.01) | 0.306 | | | | | | |
| Neuropsychology | 0.97 (0.95–0.99) | **0.006** | | | | | | |

*Note*: Bold values indicate p-values below 0.05.

Abbreviations: EQ-VAS: EuroQol visual analogue scale range; GDS-15: Geriatric Depression Scale 15; iNPH: Idiopathic normal pressure hydrocephalus; mRS: modified Rankin Scale.

EQ-VAS range 0–100 (high score higher self-rated health).

GDS-15 range 0–15 (high score more depressive symptoms).

iNPH score range 0–100 (low score more symptoms).

Radscale score range 0–12 (high score more radiological signs).

In this study, depressive symptoms (GDS-15) were three times more frequent among those with iNPH than those without iNPH. This is in line with findings in previous studies, which found a higher rate of depressive symptoms in iNPH compared to controls [10,11,26–28].

In a general population of Swedish elderly, the prevalence of suspected depression was found to be 9% [29], which is slightly lower than our figures for unlikely iNPH (14%). For numbers on depression in iNPH, both higher and lower numbers of depression in iNPH have been reported. The prevalence of suspected depression was lower in our iNPH population (27%) compared to previously reported prevalences from Finland (47%) and Sweden (46%) [10,11]. This might be because these studies included hospital-based materials with iNPH patients already referred for surgery, suggesting that these patients had worse symptoms. On the other hand, a Japanese hospital-based study showed lower rates of depression in iNPH patients (14%) [9]. Adequate comparisons between the studies are hampered by the fact that Junkkari et al.[11] and Kito et al.[9] used different assessment tools for depression than in the present study.

It is well known that iNPH causes a gradual deterioration of gait and balance, which often leads to a significant functional impairment, especially if untreated [30]. As expected, the proportion with disability (mRS ≥ 2) was more than twice as high in our iNPH population compared to those without, and they also had twice as high median scores of mRS. In preoperative clinical materials, higher rates of disability have been reported in iNPH patients [10,12,31–34], although two studies from Sweden reported mRS values more in line with ours [30,35].

**Table 6. Low EQ5D-5L index (lowest quartile of EQ5D-5L index) in relation to clinical findings, depressive symptoms, and radiological signs of iNPH.**

| Independent factors | Univariate logistic regression models (n = 119) | | Multiple logistic regression models (n = 119) | | | | | |
| | | | Model A age, sex, Radscale, iNPH score, mRS and GDS-15 | | Model B age, sex, Radscale and iNPH score | | Model C age, sex, mRS and GDS-15 | |
| | OR (CI) | P value | OR (CI) | P value | OR (CI) | P value | OR (CI) | P value |
|---|---|---|---|---|---|---|---|---|
| Age | 1.02 (0.95–1.10) | 0.642 | 0.90 (0.81–1.00) | 0.050 | 0.91 (0.82–1.00) | 0.061 | 0.94 (0.85–1.02) | 0.146 |
| Sex (male reference) | 1.30 (0.56–3.02) | 0.548 | 1.20 (0.41–3.49) | 0.736 | 1.41 (0.52–3.85) | 0.502 | 1.16 (0.42–3.18) | 0.781 |
| Radscale score | 1.30 (1.06–1.58) | **0.011** | 1.09 (0.83–1.42) | 0.548 | 1.13 (0.89–1.43) | 0.330 | | |
| modified Rankin Scale | 2.96 (1.82–4.79) | **0.000** | 1.88 (0.95–3.72) | 0.068 | | | 2.60 (1.49–4.53) | **0.001** |
| GDS-15 | 1.49 (1.22–1.82) | **0.000** | 1.24 (0.98–1.56) | 0.068 | | | 1.30 (1.04–1.61) | **0.019** |
| Total iNPH score | 0.93 (0.90–0.96) | **0.000** | 0.96 (0.91–1.01) | 0.141 | 0.92 (0.88–0.96) | **0.000** | | |
| Domain scores | | | | | | | | |
| Gait | 0.96 (0.94–0.98) | **0.000** | | | | | | |
| Balance | 0.95 (0.92–0.98) | **0.002** | | | | | | |
| Continence | 0.97 (0.96–0.99) | **0.002** | | | | | | |
| Neuropsychology | 0.97 (0.95–0.99) | **0.008** | | | | | | |

*Note*: Bold values indicate p-values below 0.05.

Abbreviations: EQ5D-5L EuroQol five-dimensions 5 levels instrument; GDS-15: Geriatric Depression Scale 15; iNPH: Idiopathic normal pressure hydrocephalus; mRS: modified Rankin Scale.

GDS-15 range 0–15 (high score more depressive symptoms).

iNPH score range 0–100 (low score more symptoms).

Radscale score range 0–12 (high score more radiological signs).

In this study, the estimated HRQoL was lower for those with iNPH compared to those without iNPH, which is in line with previous reports on this subject [10,11]. Likewise, our results that more severe iNPH symptoms, depressive symptoms, and lower functional levels were independently associated with low HRQoL correspond with findings in previous publications [11,36].

In a prospective study, Junkkari et al. [37] showed that 43% of 145 iNPH patients experienced a clinically significant improvement in HRQoL one year after shunt surgery. In the same material, the authors found that participants with a poor starting point (a higher degree of iNPH symptoms) were less likely to experience improved HRQoL despite a favourable clinical outcome [38]. iNPH patients who had to wait for surgery had a worse clinical outcome compared to those who were operated on early [30,39]. Based on these findings, iNPH patients should probably be operated on at an earlier stage to prevent disability that affects long-term HRQoL. Learning more about which factors have the largest impact on the quality of life in iNPH is probably important for optimising future treatment and care for iNPH patients.

Another topic that has not been thoroughly investigated is depression in iNPH, and the efficacy of anti-depressant drugs in iNPH has not been addressed in the literature. Symptoms that have been shown to be frequent in patients with iNPH include apathy and anxiety, which can mimic symptoms of depression or coexist in those with depressive disorders [9]. Shunt surgery might have an effect on depression in iNPH patients, which is supported by the results from a Japanese study, with a decrease in depression rates from 36% before surgery to 5% three months afterwards [12].

The cause or aetiology of depressive symptoms in iNPH is to a large extent unknown. It could be due to the same pathophysiological changes as in endogenous depression and be aggravated by the psychosocial and physical impairment associated with the iNPH disorder. However, depression in iNPH could also be an effect of the disorder itself and its specific neurobiological changes. In support for this, Peterson et al. [40], found associations between cognitive and neuropsychiatric symptoms and structural changes in patients with iNPH who underwent volumetric measurements pre and post shunting. Another mechanism of possible relevance for iNPH is the vascular depression hypothesis that may predispose or perpetuate depression among elderly [41]. According to this hypothesis, vascular changes affect the cerebral blood flow in the frontal cortex and its modulating systems, which could induce typical depressive symptoms such as lack of spontaneity, cognitive impairment, and apathy. Since cardiovascular risk factors seem to be overrepresented in iNPH [10], the vascular depression hypothesis could also contribute to depression in iNPH.

The major strength of this study is its population-based design, implying our results are applicable on all iNPH patients, not only patients diagnosed at specialised clinics. The limitations of this study are the small number of participants and the cross-sectional design, as no conclusions can be drawn regarding cause and effect. In addition, we used screening instruments to assess HRQoL as well as the number of depressive symptoms, which is not a substitution for professional clinical judgement.

## Conclusions

Our results indicate that individuals with iNPH from the general population, who have less severe symptoms compared to the clinical materials of iNPH, are still likely to have more depressive symptoms, lower functional status, and worse quality of life compared to those without iNPH. The level of functioning, iNPH symptoms, and depressive symptoms had the strongest independent association with low HRQoL.

These results underline the value of shunt surgery because of its potential to reduce symptoms and disability in iNPH and therefore improve HRQoL. Further studies are needed to determine whether shunt surgery is the most effective treatment for depressive symptoms.

## Supporting information

**S1 Fig. Participant with imaging findings of iNPH, axial view.** Ventriculomegali, Evans' index > 0.30.
(TIF)

**S2 Fig. Participant with imaging findings of iNPH, coronal view.** DESH/ Widening of Sylvian fissures, combined with narrowing of the sulci over the high convexity.
(TIF)

## Acknowledgments

We would like to thank the staff at the neurological and radiological departments at Östersund's hospital for their administrative and technical assistance. Finally, we would like to express our deepest gratitude to all the study participants for joining the study.

## Author Contributions

**Conceptualization:** Johanna Andersson, Katarina Laurell.

**Data curation:** Johanna Andersson, Mathilda Sjövill.

**Formal analysis:** Johanna Andersson, Martin Maripuu, Mathilda Sjövill, Anna Lindam, Katarina Laurell.

**Funding acquisition:** Katarina Laurell.

**Investigation:** Johanna Andersson, Martin Maripuu, Katarina Laurell.

**Methodology:** Johanna Andersson, Martin Maripuu, Katarina Laurell.

**Project administration:** Johanna Andersson, Mathilda Sjövill, Katarina Laurell.

**Resources:** Martin Maripuu, Katarina Laurell.

**Supervision:** Martin Maripuu, Anna Lindam, Katarina Laurell.

**Visualization:** Anna Lindam.

**Writing – original draft:** Johanna Andersson, Martin Maripuu, Katarina Laurell.

**Writing – review & editing:** Johanna Andersson, Martin Maripuu, Mathilda Sjövill, Anna Lindam, Katarina Laurell.

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
