## [Decision Letter · Decision Letter 0]

17 Jan 2024

PONE-D-23-34214Depressive symptoms, functional impairment, and health-related quality of life in idiopathic normal pressure hydrocephalus: a population-based studyPLOS ONE

Dear Dr. Andersson,

Thank you for submitting your manuscript to PLOS ONE. After careful consideration, we feel that it has merit but does not fully meet PLOS ONE’s publication criteria as it currently stands. Therefore, we invite you to submit a revised version of the manuscript that addresses the points raised during the review process.

Please submit your revised manuscript by Mar 02 2024 11:59PM. If you will need more time than this to complete your revisions, please reply to this message or contact the journal office at plosone@plos.org. Please include the following items when submitting your revised manuscript:A rebuttal letter that responds to each point raised by the academic editor and reviewer(s). You should upload this letter as a separate file labeled 'Response to Reviewers'.A marked-up copy of your manuscript that highlights changes made to the original version. You should upload this as a separate file labeled 'Revised Manuscript with Track Changes'.An unmarked version of your revised paper without tracked changes. You should upload this as a separate file labeled 'Manuscript'.

We look forward to receiving your revised manuscript.

Kind regards,

Jing Zhang, MD, PhD

Academic Editor

PLOS ONE

Journal Requirements:

"We would like to thank region Jämtland-Härjedalen for funding the study, the staff at the neurological and radiological departments at Östersund’s hospital for their administrative and technical assistance. Finally, we would like to express our deepest gratitude to all the study participants for joining the study."

"KL, JA, MM received governmental funding from Region Jämtland Härjedalen. JA received funding from the donation funds of "Syskonen Perssons donations fond" , "Jämtlands cancer och omvårdnads fond" and "Stiftelsen Forskningsfonden för klinisk neurovetenskap vid NUS".

4.We noted in your submission details that a portion of your manuscript may have been presented or published elsewhere. [Yes, a previous version of this article was a part of my dissertation. It has not previously been published elsewhere. Therefore it does not constitute dual publication.

Links and identification id regarding this dissertation below:

https://www.diva-portal.org/smash/record.jsf?pid=diva2%3A1611256&dswid=-1288

URN: urn:nbn:se:umu:diva-189510

ISBN: 978-91-7855-652-6 (tryckt)

ISBN: 978-91-7855-653-3 (digital)

OAI: oai:DiVA.org:umu-189510

DiVA, id: diva2:1611256] Please clarify whether this [conference proceeding or publication] was peer-reviewed and formally published. If this work was previously peer-reviewed and published, in the cover letter please provide the reason that this work does not constitute dual publication and should be included in the current manuscript.

5. In the online submission form, you indicated that [Data are available on request, provided that approval from the Regional Ethics

Committee is given. This is because it was stated in the application to the ethical

board, as well as in the information to the study participants, that data are kept

confidential and only accessible for the researchers responsible for the study.

Requests may be sent to the corresponding author or to the Regional Ethics

Committee, Umeå. The address for such a request is:

Regionala Etikprövningsnämnden i Umeå,

Samverkanshuset, C/o Umeå Universitet, 901 87

UMEÅ SWEDEN

E-mail: epn@adm.umu.se]. 

Additional Editor Comments:

While the manuscript contains crucial information on iNPH, there is room for enhancement by incorporating the feedback provided by the reviewers.

Reviewers' comments:

Reviewer's Responses to Questions

**Comments to the Author**

1. Is the manuscript technically sound, and do the data support the conclusions?

Reviewer #1: Partly

Reviewer #2: Yes

2. Has the statistical analysis been performed appropriately and rigorously? 

Reviewer #1: Yes

Reviewer #2: Yes

3. Have the authors made all data underlying the findings in their manuscript fully available?

Reviewer #1: Yes

Reviewer #2: No

4. Is the manuscript presented in an intelligible fashion and written in standard English?

Reviewer #1: Yes

Reviewer #2: Yes

5. Review Comments to the Author

Reviewer #1: The study is a population-based investigation on the impact of idiopathic normal pressure hydrocephalus (iNPH) on quality of life, depressive symptoms, and functional disability. Through neurological examinations and brain computed tomography scans of 122 participants, the researchers found that those with iNPH reported higher depression scores and lower functioning status compared to those without iNPH, and their health-related quality of life (HRQoL) was poorer. This study is interesting, but I still have some questions as follows:1. The participants in this study were recruited from a previous cohort study and followed up for two years. The authors should provide detailed inclusion and exclusion criteria for these participants, indicating that they had undergone relevant checks before being included in the study cohort to ensure that they did not have any diseases that could affect the results of this study (for example, whether the participants had excluded the possibility of depressive mood before being included in the study cohort, the authors should elaborate).2. The authors should detail when they assessed depressive symptoms, functional impairment, and/or HRQoL scores in the participants, ensuring that all scales were evaluated at approximately the same time point to exclude possible errors caused by disease progression.3. Although Tables 2 and 3 show different data, there is some duplication in the description of lines 196-199 in the results and lines 212-217. The authors should consider integrating them into one paragraph and more accurately describing the related results. 4. In Table 4, why did the authors select three different models (A, B, C) for logistic regression? The authors should give reasons; the authors should also provide detailed explanations of the differences in their analysis results to facilitate readers' understanding.

Reviewer #2: This study conducted an association analysis of anxiety symptoms, functional status and quality of life among iNPH patients in a population-based sample. In general, the sample size and analysis angle included in this article were acceptable. However, the following aspects need to be further improved by the author:

1. Please clarify the grouping criteria in detail. How to clearly define "iNPH patients" and "unlikely iNPH patients".

2. All participants underwent head imaging, and I recommend supplementating with representative imaging images.

3. The study sample is the population of patients enrolled in 2014 and followed up in 2017. Whether the authors can further follow-up up to 2024 makes the conclusion more convincing.There may be some new discoveries.

6. PLOS authors have the option to publish the peer review history of their article (what does this mean?). If published, this will include your full peer review and any attached files.

Reviewer #1: No

Reviewer #2: No

---

## [Author Response · Author response to Decision Letter 0]

1 May 2024

Dear Editors and Reviewers,

We sincerely appreciate your time and insightful comments, which we believe have significantly enhanced the quality of the manuscript.

We have addressed your points in the document titled "Response to Reviewers" and made revisions accordingly. 

Additionally, we have updated the funding statement and provided further clarification regarding the concern of dual publication in the revised cover letter.

Thank you once again for your valuable input and consideration.

Best regards, Johanna Andersson

---

## [Decision Letter · Decision Letter 1]

17 Jul 2024

Depressive symptoms, functional impairment, and health-related quality of life in idiopathic normal pressure hydrocephalus: a population-based study

PONE-D-23-34214R1

Dear Dr. Andersson,

We’re pleased to inform you that your manuscript has been judged scientifically suitable for publication and will be formally accepted for publication once it meets all outstanding technical requirements.

Kind regards,

Jing A. Zhang, MD, PhD

Academic Editor

PLOS ONE

Additional Editor Comments (optional):

Reviewers' comments:

Reviewer's Responses to Questions

**Comments to the Author**

1. If the authors have adequately addressed your comments raised in a previous round of review and you feel that this manuscript is now acceptable for publication, you may indicate that here to bypass the “Comments to the Author” section, enter your conflict of interest statement in the “Confidential to Editor” section, and submit your "Accept" recommendation.

Reviewer #1: All comments have been addressed

Reviewer #2: All comments have been addressed

2. Is the manuscript technically sound, and do the data support the conclusions?

Reviewer #1: Yes

Reviewer #2: Partly

3. Has the statistical analysis been performed appropriately and rigorously? 

Reviewer #1: Yes

Reviewer #2: Yes

4. Have the authors made all data underlying the findings in their manuscript fully available?

Reviewer #1: Yes

Reviewer #2: Yes

5. Is the manuscript presented in an intelligible fashion and written in standard English?

Reviewer #1: Yes

Reviewer #2: Yes

6. Review Comments to the Author

Reviewer #1: (No Response)

Reviewer #2: The authors have revised and responded well to the issues raised, making the article more logical and the conclusions more convincing. I recommend acceptance of the article for publication.

7. PLOS authors have the option to publish the peer review history of their article (what does this mean?). If published, this will include your full peer review and any attached files.

Reviewer #1: No

Reviewer #2: No

---

## [Editor Report · Acceptance letter]

22 Jul 2024

PONE-D-23-34214R1 

PLOS ONE

Dear Dr. Andersson, 

I'm pleased to inform you that your manuscript has been deemed suitable for publication in PLOS ONE. Congratulations! Your manuscript is now being handed over to our production team.

Kind regards, 

on behalf of

Dr. Jing A. Zhang 

Academic Editor

PLOS ONE